# Local Fractions - a method for the calculation of local source contributions to air pollution, illustrated by examples using the EMEP MSC-W model (rv4_33)

Peter Wind[1,2], Bruce Rolstad Denby[1], and Michael Gauss[1]

[1]Climate Modelling and Air Pollution Division, Research and Development Department, Norwegian Meteorological Institute (MET Norway), PO 43 Blindern, 0313 Oslo, Norway
[2]Department of Chemistry, UiT - The Arctic University of Norway, N-9037 Tromsø, Norway

**Correspondence:** Peter Wind (peter.wind@met.no)

**Abstract.** We present a computationally inexpensive method for individually quantifying the contributions from different sources to local air pollution. It can explicitly distinguish between regional/background and local/urban air pollution, allowing fully consistent downscaling schemes.

The method can be implemented in existing Eulerian chemical transport models and can be used to distinguish the contribution of a large number of emission sources to air pollution in every receptor grid cell within one single model simulation and thus to provide detailed maps of the origin of the pollutants. Hence it can be used for time-critical operational services providing scientific information as input to local policy decisions on air pollution abatement. The main limitation in its current version is that non-linear chemical processes are not accounted for and only primary pollutants can be addressed.

In this paper we provide a technical description of the method and discuss various applications for scientific and policy purposes.

## 1 Introduction

The origin of atmospheric pollutants within a given region is one of the fundamental questions of air quality research. Degradation of air quality, either temporary or sustained, is often the result of both local and long-range transported air pollution, originating from anthropogenic but also natural emission sources. Anthropogenic emissions are due to a large number of different categories such as road traffic, industrial point sources and large area sources.

In order to devise optimal strategies of air pollution abatement, for example short-term or long-term emission reduction measures, air quality managers need to have access to reliable scientific knowledge about the origin of air pollution. Typical questions include: a) By what amount can local air pollution be reduced through local measures only, and in which cases will regional or countrywide measures be necessary? b) What will be the benefit of emission reduction measures imposed on one or several specific emission sectors? c) Will these measures be efficient on a short time frame or should they be implemented on a longer-term basis?

Many different methods exist to extract information about the origin of air pollution (e.g., Thunis et al. , 2019; Clappier et al., 2017). Some of them are based on measurements of the chemical composition of air masses in the region or interest (receptor region). Such a 'chemical fingerprint' can then give hints on the origin (source region or sector) of the pollutants. Most methods, however, are based on models as these can be readily applied to scenario calculations as well. Chemical Transport Models (CTMs), in particular, are efficient mathematical tools that treat emission sources, transport, chemical conversion and loss mechanisms of air pollutants in a consistent way, and allow different scenarios to be assessed within a reasonable amount of computing time.

The simplest method to evaluate the importance of different emission sources in a CTM is the 'direct' method (e.g., Folberth et al., 2012), sometimes also referred to as 'annihilation' or 'brute force' method, where the same model simulation is repeated with and without including a chosen emission source. The difference in pollutant concentrations in the receptor region can then be attributed to the chosen emission source (impact). In order to stay within quasi-linearity one can choose to reduce the emission source by only a small amount. This is usually referred to as 'perturbation method' (e.g., Jacob et al., 1999; Fiore et al., 2009) and is well suited to simulate the effects of policy measures to reduce emissions from certain sectors by a certain amount. However, one of the drawbacks of this method is that for each source contribution a new, independent, simulation must be performed.

Many chemical processes in the atmosphere are non-linear. For example, a doubling of the emission from one specific source will not necessarily double its contribution to air pollution levels. This also implies that the sum of contributions (from individual sources) calculated by the direct method (or by perturbation methods) will in general not be equal to the total air pollution level calculated in a simulation where emissions from all sources are included in full. Consequently, one has to distinguish between two different questions: 1) What is the effect of a change in emissions from individual sources on air pollution? (air pollution sensitivity), and 2) What are the contributions of individual sources to air pollution? (source apportionment). Due to non-linearities, question 2 cannot be answered by reducing the emissions of individual sources to zero one-by-one. An alternative approach to estimate contributions from individual sources in model calculations is a technique known as "tagging", which distinguishes chemically identical molecules according to their sources. In the calculation the molecule is labelled (e.g. by a separate index) according to its source and then keeps this label during transport and chemical transformations. When analyzing air pollution levels within a given receptor area, the fractions of molecules with different labels can be considered separately, thereby giving an estimate of the contributions from the different sources. A series of methods have been proposed to address the contribution from different sources based on the 'tagging method' (e.g., Butler et al., 2018; Emmons et al., 2012; Dunker et al. , 2002; Kwok et al. , 2015; Grewe et al., 2013, 2017; Wang et al., 1998; Wu et al., 2011).

Tagging methods are also useful for tracing primary pollutants (e.g., Kranenburg et al., 2013). However, in cases where the number of different tagged sources is large, the tagging methods can become excessively computationally expensive.

In this regard, 'adjoint models' (Elbern and Schmidt, 1999; Vautard et al., 2000; Henze et al., 2007) are superior. Adjoint models calculate the derivative of a model scalar with respect to all other model parameters in one single simulation and in this

way efficiently quantify the contribution from all emission sources to air pollution in a given receptor region. However, a new adjoint simulation must be performed for each receptor region.

Still, only a relatively small amount of sources or receptors at a time can be analyzed by all these methods. Perturbation methods calculate all receptor values for one source group, tagging methods compute all receptors for a limited number of source groups, while adjoint models address all sources for one receptor group. Ideally, all contributions to all receptor points should be described.

In this paper we present a method which can efficiently calculate the contribution of a significantly larger amount of sources (thousands or more), to a limited (but large) number of receptor regions. This method does not provide results that cannot be obtained by other means, but it does so at a lower computational cost and is thus well suited especially for time critical operational applications. It can be built on top of existing Eulerian CTMs relatively easily, and thereby has the potential to offer a new range of applications.

An important limitation is that the method is limited to primary pollutants, for which linearity can be assumed. It will thus complement existing methods, but not replace them.

In principle the method allows the definition of any group of sources, but here we will show results only for the case where each defined source is defined within a single grid cell. One key limitation, which makes the method manageable, is that the tagged values are stored only up to a preset horizontal and vertical distance from their source. We will call the region within this distance the "local region". The size of this region must be set as a balance between computational cost and the accuracy requirement of the application.

In the following Section we describe the method in technical detail, while in Sect. 3 we show concrete examples of what kind of results the method can provide, and how to quantify some of the limitations associated with the method. The results will also be compared against the direct method. In Sect. 4 we will give an overview over what is required to implement the method in an existing CTM and discuss the performance in the EMEP MSC-W implementation. Finally in the last section we discuss possible applications of the method as well as plans for further development.

## 2  Description of the method

In theory the method corresponds to a tagging method, where pollutants from different origins are tagged and their values are traced and stored individually. However the total amount of pollutants is not computed as a sum of tagged values; instead the tagged values show which fraction of the total pollutants originate from a specific origin.

We define the Local Fraction $LF_s$ in a receptor grid cell as the fraction of pollutant that is due to a particular source term $s$. $s$ is the index defining a pollutant or a pollutant from a specific sector. For example, $s$ can refer to primary particulate matter from any sector, or restricted to a power plant or the road traffic emissions in a specific source region. $LF_s$ is a number between zero and one and is calculated as:

$$LF_s = \frac{\text{Pollutant due to source } s}{\text{Total Pollutant}} = \frac{LP_s}{TP} \tag{1}$$

The Total Pollutant is abbreviated $TP$; it could be the air concentration of particulate matter for example. The Local Pollutant, $LP_s$, is the part of $TP$ "tagged" from a specific origin $s$. Its value is in general the result of various processes (emissions, advection, diffusion, etc.) as will be described below. Given the value of the Total Pollutant, then the Local Fraction and the Local Pollutant carry the same information, but as we will see, there are a few practical advantages of storing the Local Fraction rather than the Local Pollutant.

In a time-splitting framework the different physical processes are included sequentially, and we will show in the next Sections how the value of the Local Pollutant changes during each of them. For simplicity, the initial value for $LF_s$ is set to zero, given that in the long term $LF_s$ should not be sensitive to the initial value.

## 2.1 Emissions

The Local Pollutant and Local Fraction are associated with a particular emission source category ($E_s$) in a specific grid cell. (Formally the source could also be spread over a group of grid cells, but at present we limit ourselves to sources defined on single grid cells). If $E_s(t)$ is the emission rate of source $s$ at time $t$, $LP_s$ will increase during the time step $\Delta t$:

$$LP_s(t + \Delta t) = LP_s(t) + E_s(t)\Delta t \tag{2}$$

and

$$LF_s(t + \Delta t) = \frac{LP_s(t + \Delta t)}{TP(t + \Delta t)} \tag{3}$$

For instance $s$ could refer to emissions of particulate matter from road traffic emissions, $TP$ would be the total concentration of particulate matter in the receptor region, and $LF_s$ would then be the fraction by which the total concentrations in the receptor region would be reduced if the emissions from road traffic in the source region were removed completely (assuming linearity).

## 2.2 Advection

Transport of pollutants will mix pollutants from different origins. We will trace individually the Local Pollutant due to different sources and from every horizontal grid cell within the source region. We need then two sets of position indices, one for the origin (source region) and one for the actual position (receptor grid cell):

$$LF_{s,x_s,y_s}(x, y, z, t) \tag{4}$$

Where $x_s$ and $y_s$ are the (horizontal) coordinates of the source grid cell, and $x$, $y$ and $z$ are the coordinates of the receptor grid cell. $s$ is a specific source category at $(x_s, y_s)$. In order to keep the calculation at a reasonable cost, one can limit $x_s$ and $y_s$ to be within a preset number of grid cells from the receptor grid cell, $\Delta^{max}$:

$$x - \Delta^{max} < x_s < x + \Delta^{max} \qquad\qquad y - \Delta^{max} < y_s < y + \Delta^{max} \tag{5}$$

The source position indices are then replaced by its relative position relative to the receptor grid cell:

$$LF_{s,\Delta x_s, \Delta y_s}(x,y,z,t) \qquad (6)$$

where $\Delta x_s = x_s - x$ and $\Delta y_s = y_s - y$ are the signed distances to the source. In practice, also z is limited, as it is usually not necessary to trace pollutants for receptor grid cells all the way up through the atmosphere. Note that the vertical position of the source is not explicitly traced, but it can, in principle, be included in the form of separate sources $s$.

We call the region delimited by all $(x_s, y_s)$ and the vertical range of $z$ for the "local region".

$LF_{s,\Delta x_s, \Delta y_s}(x,y,z,t)$ is in practice a seven dimensional array. The range of $s$ depends on the number of source categories
to be tracked. The size of this array can be very large, which reflects the large amount of information it carries.

Pollutants can be traced within this region. If they leave the local region, they are no longer identifiable by the method, even if they return into the local region.

Let us consider a flux of pollutant, $F(x,y,z,t)$ (assumed positive), from a grid cell $x$ to $x+1$ during $\Delta t$, and a source at a position $\Delta x_s$ relative to x.

The amount of Local Pollutant leaving the grid cell $x$ is

$$F(x,y,z,t)LF_{s,\Delta x_s, \Delta y_s}(x,y,z,t) \qquad (7)$$

At position $x+1$, the relative position of that source is $x_s - 1$, and the Local Pollutant is thus updated according to

$$LP_{s,\Delta x_s - 1, \Delta y_s}(x+1,y,z,t+\Delta t) = LP_{s,\Delta x_s - 1, \Delta y_s}(x+1,y,z,t) + F(x,y,z,t)LF_{s,\Delta x_s, \Delta y_s}(x,y,z,t) \qquad (8)$$

Or, if the source is moved by one grid cell ($\Delta x_s$ replaced by $\Delta x_s + 1$), the formula becomes:

$$LP_{s,\Delta x_s, \Delta y_s}(x+1,y,z,t+\Delta t) = LP_{s,\Delta x_s, \Delta y_s}(x+1,y,z,t) + F(x,y,z,t)LF_{s,\Delta x_s + 1, \Delta y_s}(x,y,z,t) \qquad (9)$$

The Local Fractions are then updated according to the definition in Eq. (1).

$$LF_{s,\Delta x_s + 1, \Delta y_s}(x+1,y,z,t+\Delta t) = \frac{LP_{s,\Delta x_s + 1, \Delta y_s}(x+1,y,z,t+\Delta t)}{TP(x+1,y,z,t+\Delta t)} \qquad (10)$$

The fluxes and Total Pollutants are not explicitly dependent on the source $s$, and are normally available quantities in the CTM model.

If the flux is exiting the grid cell $x$, the Local Fractions at $x$ do not have to be updated, since it can be assumed that the fractions being removed are the same for the Local and Total Pollutants.

## 2.3 Diffusion (and convection)

For diffusion we compute the effect of diffusion directly on every Local Pollutant:

$$LF_{s,\Delta x_s,\Delta y_s}(x,y,:,t+\Delta t) = \frac{\text{Diffusion}(LP_{s,\Delta x_s,\Delta y_s}(x,y,:,t))}{\text{Diffusion}(TP(x,y,:,t))} \tag{11}$$

Where "Diffusion()" is the numerical operator that computes the diffusion in the model and the colon ':' indicates its operation over the entire vertical grid column. This ensures a consistent treatment of the Local Fractions, whatever numerical procedure is applied for the diffusion.

In a practical implementation it is not necessary to include all the vertical levels, as the contribution from higher levels is negligible (it corresponds to pollutants leaving and returning to the local region during the same time step). In our implementation we include only two layers above the highest local region considered.

For convection the same procedure can be used by replacing the diffusion operator in Eq. (11) by the convection operator . In the current EMEP MSC-W model version, the convective processes are not implemented in the Local Fraction calculations.

## 2.4 Deposition

For deposition (dry or wet), we can assume that the same fractions of Local and Total Pollutants are removed. Therefore the Local Fraction will not vary during the deposition process:

$$LF(t+\Delta t) = LF(t) \tag{12}$$

The simplicity of this formula is one of the motivations for storing $LF$ rather than $LP$.

## 2.5 Chemistry

To fully follow the pollutants through all the chemical reactions would, in principle, require an explicit reference to all the sources and grids. It is possible to reduce the size of the problem if linearity is assumed. This has been done by other groups (e.g., Kranenburg et al., 2013). The calculation of all the chemical reactions is one of the most computationally intensive part of CTMs (roughly 60% in the EMEP MSC-W model (Simpson et al. , 2012) used for the tests presented below). A consistent chemical treatment of Local Pollutants would mean to almost multiply the computation time by the number of Local Pollutants considered, i.e. the size of $(s, \Delta x_s, \Delta y_s)$. In order to preserve the simplicity of the method, we will in this version assume that the chemical processes modify the local and non-local part of the pollutants in the same proportions. With this assumption Eq. (12) can be used. This assumption is correct for primary particles and, as illustrated in our examples below, can give meaningful results also for $NH_3$, $SO_x$ and $NO_x$. So far the method is only developed for emitted pollutants, and not for secondary pollutants.

## 3 Examples and validation

The Local Fractions will depend on a broad range of factors such as emission distributions, meteorological conditions, grid resolution, chemical regime, size of the local region etc. It is beyond the scope of this article to systematically quantify how all the possible situations affect the Local Fractions. The limitations of the method should be estimated for each concrete application. The examples in this section also provide methods for estimating different errors associated with the method

(limitation of the size of the local region, non-linearities).

The Local Fraction $LF_{s,\Delta x_s,\Delta y_s}(x,y,z,t)$ is a 7-dimensional array, and in the following Sections we will try to briefly illustrate the information that can be provided by this array.

The results shown in this Section are based on a grid with a resolution of 0.3°in the longitude direction and 0.2°in the latitude direction. The parameter settings are essentially the same as what is used for the official EMEP MSC-W model runs, using

"TNO_MACC-III" emissions (2015 update of (Kuenen et al., 2014)). However to simplify the interpretation of the results, two important modifications have been introduced: a simplified advection scheme is used (see Sec. 3.2), and all emissions are released at the lowest level. The standard settings of the model do not include convection over Europe.

### 3.1  Time and space dependence ($\Delta x_s = \Delta y_s = 0$)

In Fig. 1, an illustration of the time evolution of the instantaneous Local Fraction for fine particulate matter ($PM_{2.5}$) at an

arbitrary location (in the Oslo agglomeration) is shown. The value gives the fraction of $PM_{2.5}$ which has its origin in the same grid cell. It is strongly correlated with the concentrations of $PM_{2.5}$, but it does not always vary exactly in the same way. It will also depend on the wind speed, emission rates and the surrounding levels of pollution. If a relatively large amount of clean air is moving into that area, the total concentration will decrease, but the Local Fraction will remain high. High Local Fractions indicate that most of the pollutant is locally produced.

Figure 2 shows a map of monthly-mean Local Fractions for March 2016. It gives a picture of how much the sources in a particular grid cell contribute compared to the surrounding sources. The distribution is similar to the emission distribution, but isolated emission sources show up more clearly in the Local Fractions map.

### 3.2  Illustration of Source Receptor capabilities

For a fixed value of $x$, $y$ and $z$ and $t$, the Local Fractions $LF_{s,\Delta x_s,\Delta y_s}(x,y,z,t)$ give the contributions of a pollutant $s$ emitted at $(x+\Delta x_s, y+\Delta y_s)$ to the position $(x,y)$, i.e. a two dimensional map of the origin of the pollutants found at position $(x,y)$. Thus provides a complete description of all source receptor relationships within a given distance from the receptor grid cell.

Figure 3 shows such a map for an arbitrary location. It is simply the value of $LF_{s,\Delta x_s,\Delta y_s}(x,y,z,t)$ averaged over one

5    month, where $x$ and $y$ are the position of the central point (receptor). Such a map is calculated for any point on the grid in a single simulation. In this example the local region has a horizontal extend of 41 times 41 grid cells. Direct methods would then, in principle, require $41 \cdot 41 + 1 = 1682$ simulations to calculate the values of one of those maps.

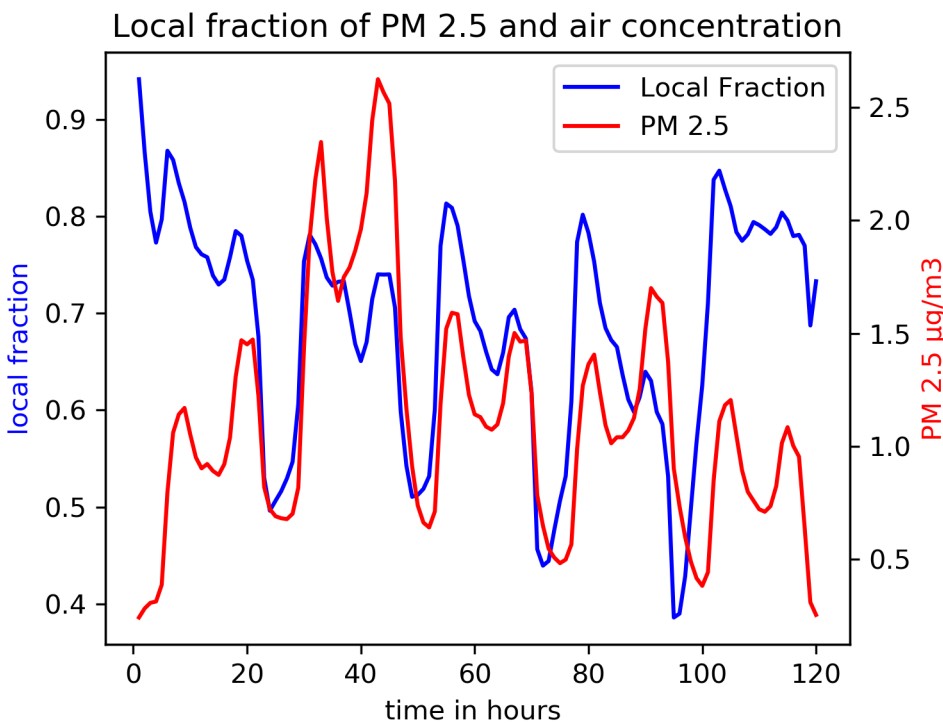

**Figure 1.** Time evolution of the Local Fraction of $PM_{2.5}$ in the Oslo agglomeration (left axis) during the period 5th to 9th January 2016 (longitude=11.55°, latitude=59.9°). The total concentration of PM is also shown (right axis).

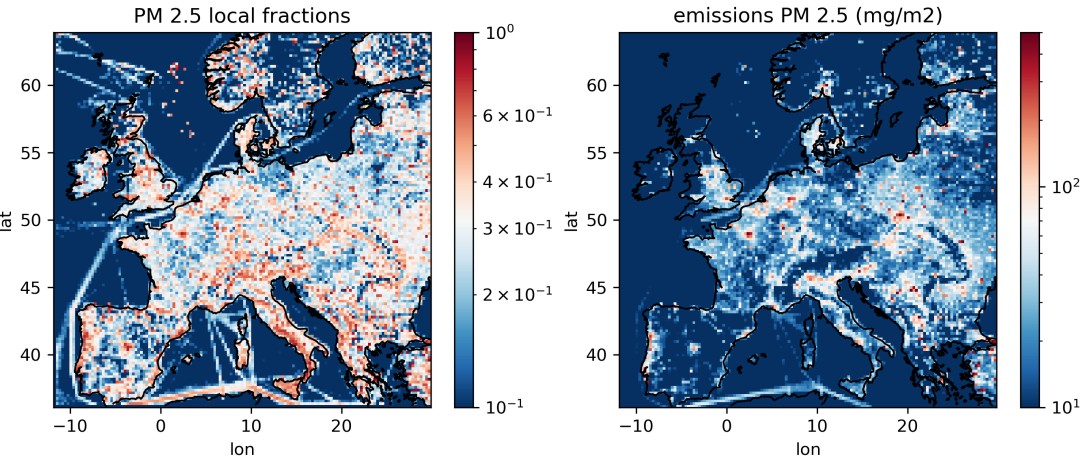

**Figure 2.** Example of spatial distribution of the Local Fraction of $PM_{2.5}$, averaged over one month (March 2016, left panel). The total emissions of $PM_{2.5}$ accumulated over that period are shown in the right panel.

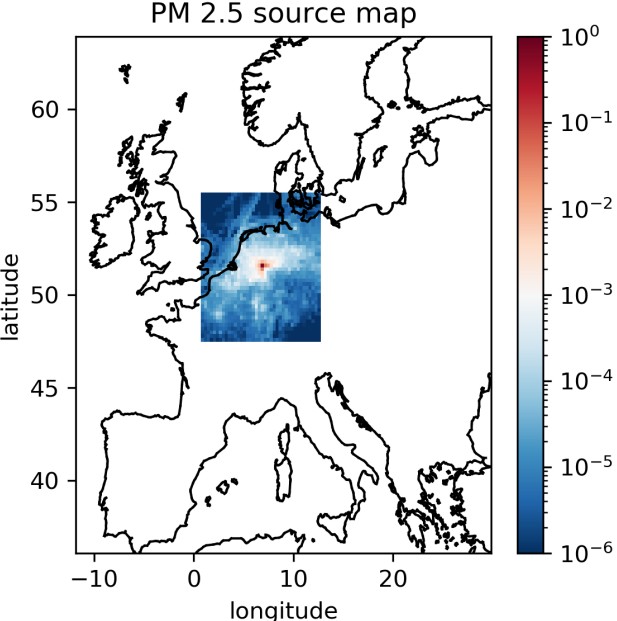

**Figure 3.** Example of Local Fractions as a source map. The values show the fraction of PM$_{2.5}$ that has been emitted at that location and transported to the central point. The sum of all the fractions is in this case 0.976, meaning that 2.4 % of the PM$_{2.5}$ concentration at the central position, originates from sources outside of the local region

.

In order to compare with the direct method, one can "invert" $LP_{s,\Delta x_s,\Delta y_s}(x,y,z,t)$, to get a map of the receptors for a fixed source:

$$LP^{\dagger}_{s,\Delta x_s,\Delta y_s}(x,y,z,t) = LP_{s,-\Delta x_s,-\Delta y_s}(x+\Delta x_s,y+\Delta y_s,z,t) \tag{13}$$

$LP^{\dagger}_{s,\Delta x_s,\Delta y_s}(x,y,z,t)$ then gives the contributions of a pollutant $s$ located at $(x,y)$ to the position $(x+\Delta x_s,y+\Delta y_s)$

Figure 4 illustrates a comparison of the results obtained

1) by removing the emissions from a single grid cell and computing the difference with the normal case (direct method).

2) by using one single run and Eq. (13) with a local region of size 41×41×8.

Within the local region the results are similar, but the Local Fraction method gives such a map for any grid cell in one single run, while the direct method would require a separate run for each source region.

Note that for the purposes of this experiment we have chosen a zero order advection scheme in all model runs. The default

5   fourth order scheme is slightly non-local, and the direct method would give spurious results very close to the sources; tracking and direct methods would give different results. For example, in the fourth order scheme, if emission are *reduced* in one gridcell, this can reduce the flux from the neighbouring grid cell in the upwind direction, thereby *increasing* the concentration

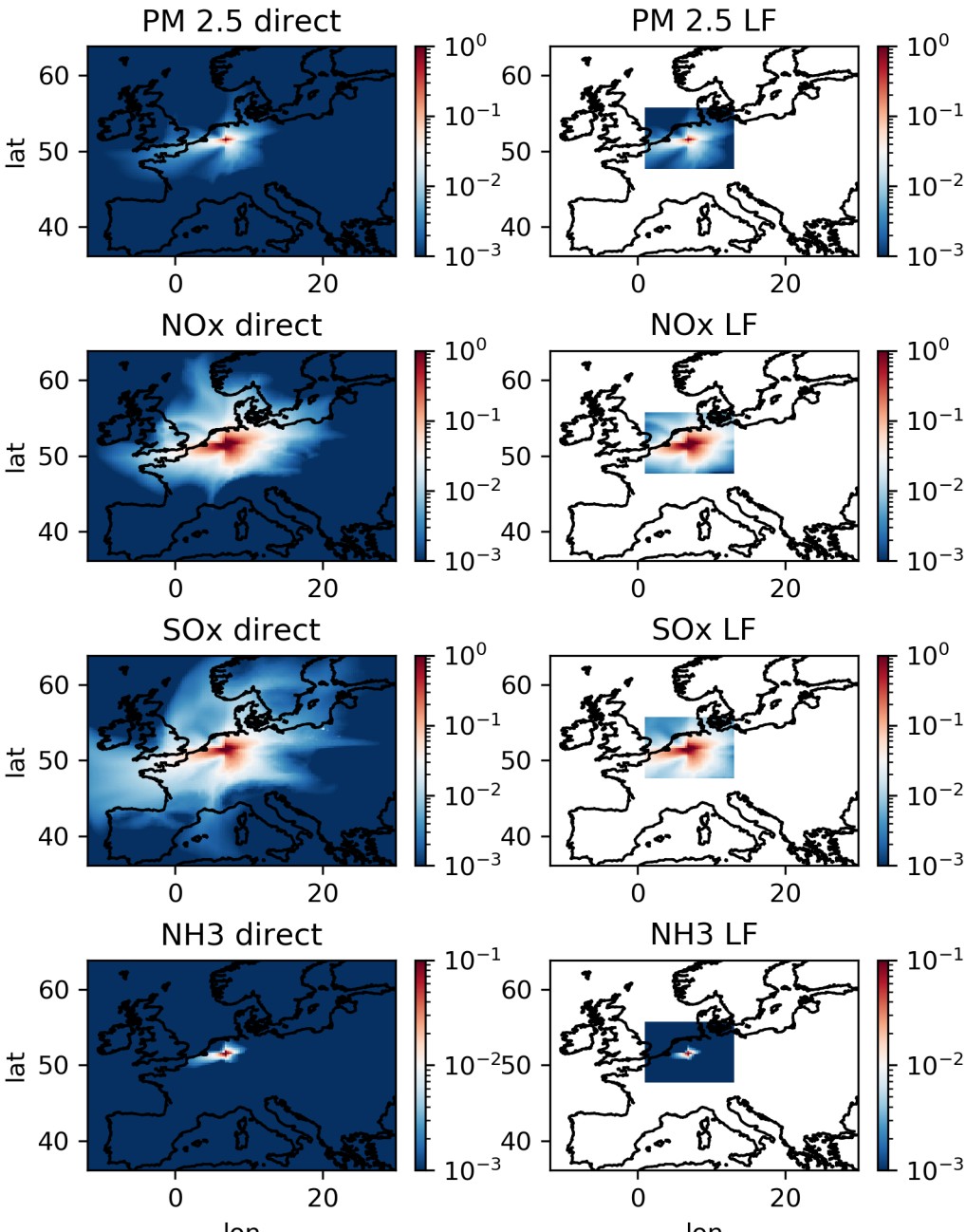

**Figure 4.** Receptor map for a single grid cell emission, obtained through direct method (left panels) and the Local Fraction method (right panels), averaged over one month (March 2016). Concentrations of $PM_{2.5}$, $NO_x$, $SO_x$ and $NH_3$ (in $\mu g\ m^{-3}$). The direct method requires a separate run for each source location. The Local Fraction method gives the receptor map in one single run, in a limited region around the source, but for any source grid cell.

of pollutants in the upwind grid cell. This is however not a problem for the LF method (or any tracking method), and for short distances it is actually an advantage compared to the direct method.

## 3.3 Vertical transport

For source apportionment applications, the focus is typically on horizontal transport. Nevertheless the code should trace the pollutants with a combination of vertical and horizontal transport. Over short distances only transport through the lowest layers needs to be considered. If the focus is on regions where a large part of the pollutants are transported over long distances, the vertical extend of the local area should be chosen large enough.

Figure 5 illustrates the dependence of the Local Fraction on the thickness of the local region. In this example, only a few vertical levels are required to describe the Local Fraction within the grid cell (the remaining discrepancy comes from pollutants first leaving the grid cell, and then returning later). For a distance of up to 14 grid cells, including 8 vertical layers in the local region, results are not distinguishable from the exact value calculated by the direct method. Obviously, emission or vertical mixing at higher altitudes would require to include the corresponding vertical layers.

For NOx, even for relatively small distances, there is a discrepancy between the contribution calculated with the Local Fraction method and the direct method (Fig. 6). This is because the Local Fraction method does not explicitly distinguish between NO and NO2. The mix modelled in the remote emissions may differ from the local values. Since reaction rates are different for NO and NO2, the local NOx transformation rate is not representative for the reaction rates of the incoming "older" NOx.

## 3.4 Completeness

For local regions that are large enough, the source of all primary particles can be accounted for. This can be verified directly by summing all the Local Fractions for a given grid cell:

$$\sum_{\Delta x_s, \Delta y_s} LF_{s, \Delta x_s, \Delta y_s}(x, y) \tag{14}$$

A sum of one means that all sources are accounted for. The difference between the sum of of the Local Fractions and one gives the fraction of pollutants with sources outside of the local region. In Fig. 7 the sum of the Local Fractions is shown for every grid cell on the map for different horizontal sizes of the local region. For most land areas, more than 80% of the sources are found for the smallest window (41x41) and essentially all sources for the largest (161x161).

Figure 8 show the result for different vertical extend of the local region. The Local Fractions get close to complete in most places, when 8 vertical levels are included (approximately 1522 meters height). As one would expect, this roughly corresponds to the maximum height of the boundary layer in March over land in Europe.

Note that incomplete results are not a measure of an error in the method. Rather they show the amount of pollutants with sources outside of the local region, which is useful information.

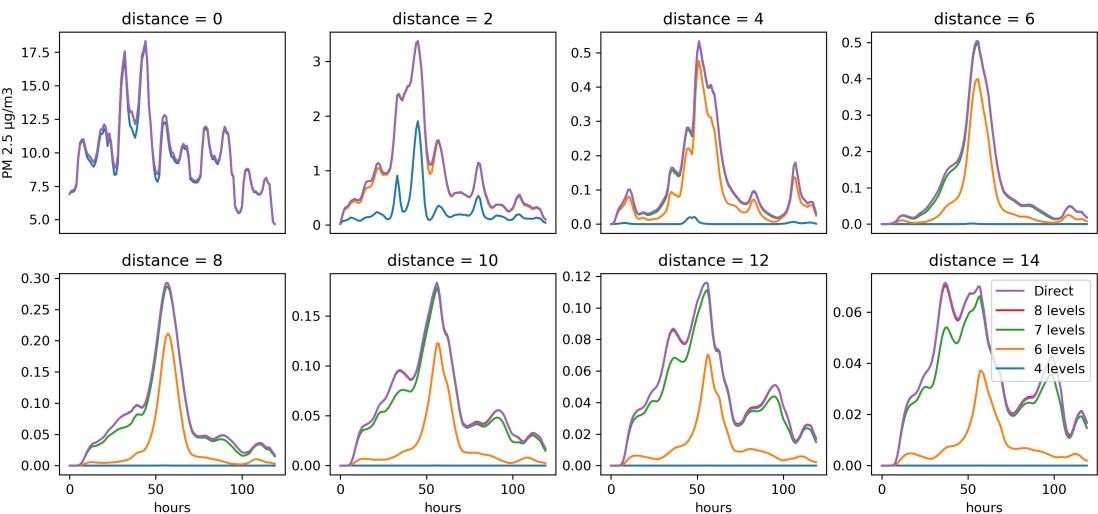

**Figure 5.** Sensitivity of the concentration of PM$_{2.5}$ ($\mu$g m$^{-3}$) to the number of vertical levels included in the local region, for different distances from the source. The distance from the source is given in numbers of grid cells (one grid cell = 0.3 degrees in longitude direction). Source in Oslo agglomeration. Horizontal axis is time (120 hours). The 8 levels results cannot be distinguished from the direct method on the figure, even for the largest distance considered.

## 4 Implementation and Computational aspects

From an implementation point of view the method is a "diagnostic" calculation, in the sense that it gives additional information extracted from existing data, in opposition to a modification of the method for computation of the concentrations of air pollutants. Therefore the method can be implemented on top of existing CTM, without having to rewrite the code for the main processes. What is required is to include calls to new routines that can perform the operations described in Sect. 2. Concretely, the main changes to be made are:

– Define the instantaneous Local Fraction 6-dimensional array $LF(s, \Delta x_s, \Delta y_s, x, y, z)$ , and one corresponding array for each of the time averaged periods (at least one for averaging over the run, and possibly another for averaging over hours for example).

– Write a routine that performs the operations from Sect. 2. In addition a routine for writing out the results (i.e. six dimensional Local Fraction arrays) and one routine should do the averaging over time.

– As input for those routine, the main code must make available the emission rates of the relevant sectors and the advection fluxes. If the fluxes are not available, or in a simplified version, the fluxes could be defined directly by an other method. For example an already good approximation would be to take $F_x = \frac{c\Delta t}{\Delta x} LP_{up}$ , where $c$ is the wind speed in x direction, $\Delta x$ the size of the gridcell and $LP_{up}$ the concentrations in the upwind gridcell.

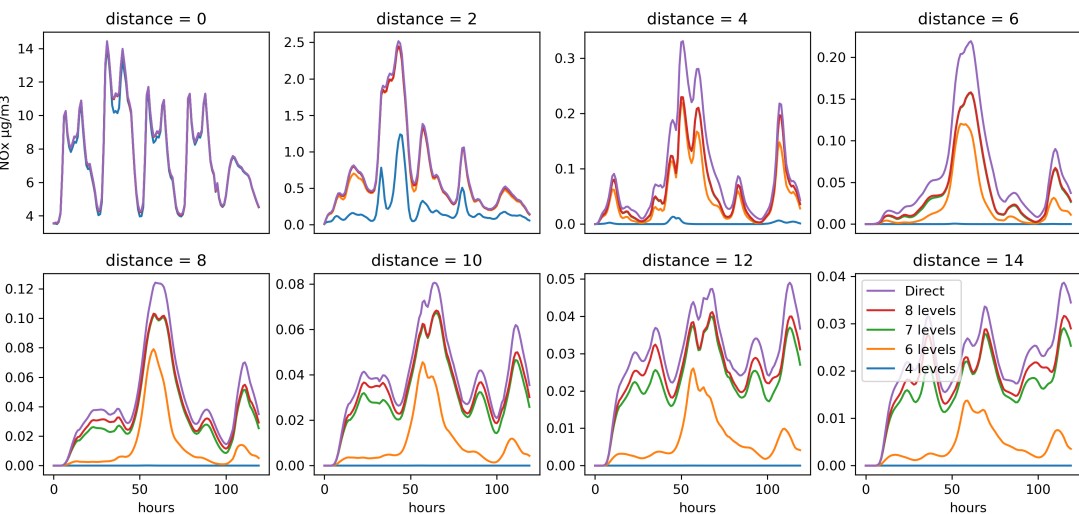

**Figure 6.** Concentration of NOx ( $\mu$g m$^{-3}$ ). Sensibility to number of vertical levels included in the local region. Distance to source in number of grid cells (one grid cell = 0.3 degrees in longitude direction). Source in Oslo agglomeration. Horizontal axis is time (120 hours). At a distances larger than a few grid cells, a discrepancy can be observed between the contribution calculated with the Local Fraction method and the direct method.

- In Eq. 9 it is necessary to have access to values from the nearest neighbour gridcells. In a parallel implementations, this
  5     may require supplementary communication routines.

- In addition, of course, the calls to those new routines have to be integrated into the main code. Also switches to choose the pollutants and the sizes of the local region have to be created.

There is no feedback of the LF calculations to the concentrations of air pollutants; those will be unaffected by the new routines. This clear separation greatly simplifies the practical development.

10     In the EMEP MSC-W implementation (rv4_33), all the extra routines are put in a separate file ("uEMEP_mod.f90"), except for the LF communication routine. If no LF output is required, those routine are not used at all, and if the LF routine are called, the rest of the code still performs exactly the same operations.

Since one of the key advantages of the Local Fraction method is its low computational demand, we will give a few concrete examples of the computational cost for providing the Local Fraction values in our implementation. The transformations carried
15     out for the calculation of Local Fractions presented in Sect. 2 are all relatively simple. The most computationally intensive parts of the model (calculation of fluxes, chemical transformations, deposition processes) are not explicitly performed for every Local Pollutant, but only once for the total concentrations. For processes were local pollutants are transformed by the same relative amount as non-local pollutants (deposition and chemistry in our implementation), there is no need to update the Local Fractions; this is the main motivation for storing the Local Fractions rather than the local pollutants.

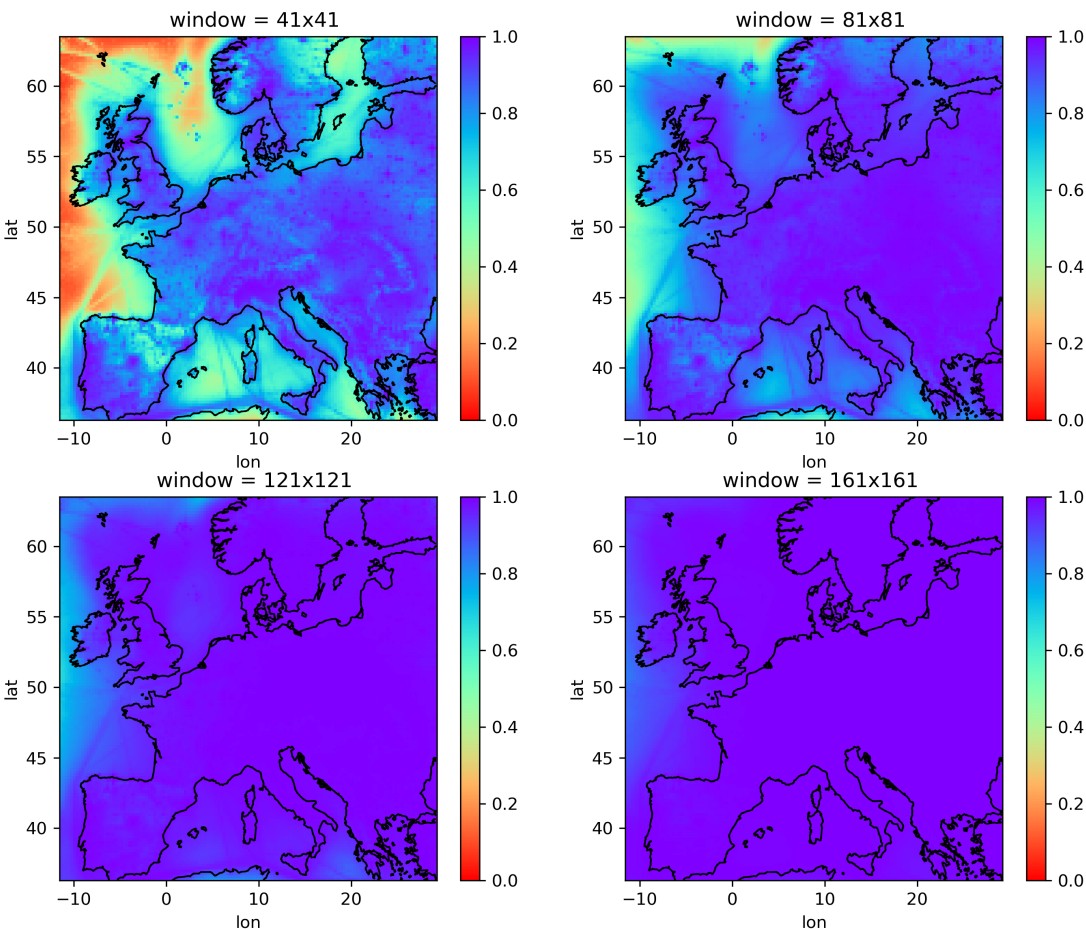

**Figure 7.** Sum of all Local Fractions (Eq. (14)) for PM$_{2.5}$ and different sizes of the local region (average for March 2016). The distance is counted as number of grid cells in each direction. All vertical layers (20) are included. A sum of 1.0 means that all the sources have been accounted for.

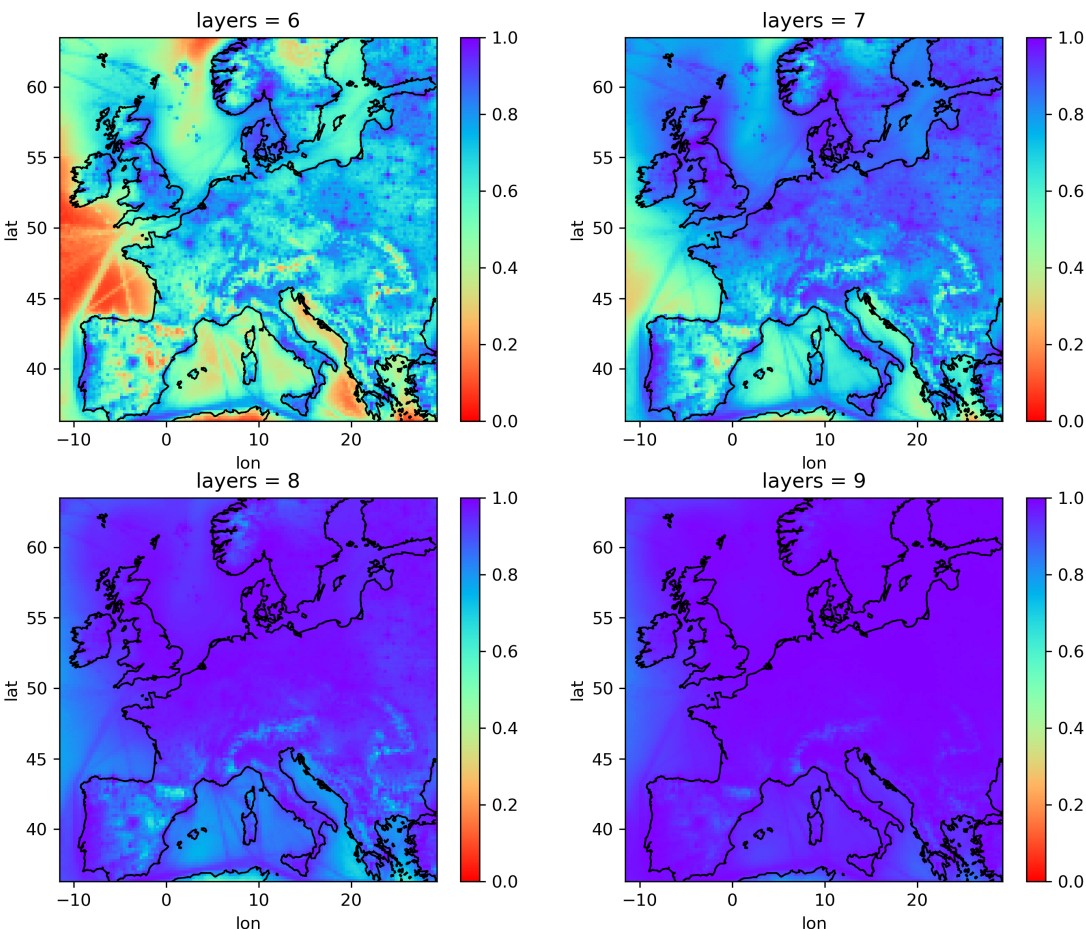

**Figure 8.** Sum of all Local Fractions (Eq. (14)) for PM$_{2.5}$ and for different vertical extents of the local region (average for March 2016). A horizontal region of 161×161 is included in the local region. For a standard atmosphere, the height of the top of the layers 6, 7, 8 and 9 are, respectively, 623, 1015, 1522 and 2149 meters. A sum of 1.0 means that all the sources have been accounted for.

| local region | total time | emission | advection | diffusion | write | averaging | comm. | other | memory |
|---|---|---|---|---|---|---|---|---|---|
| 11x11x3 | 1.9 % | 0.5 | 0.8 | 0.2 | 0.4 | 0.1 | 0.3 | -0.4 | 0.6 GB |
| 21x21x6 | 11.9 | 0.6 | 5.3 | 1.3 | 1.2 | 0.5 | 2.4 | 0.7 | 4 GB |
| 41x41x6 | 38.5 | 0.9 | 18.4 | 4.5 | 3.8 | 1.6 | 9.4 | -0.0 | 16 GB |
| 41x41x10 | 63.0 | 0.9 | 31.6 | 6.9 | 3.8 | 2.9 | 17.1 | -0.3 | 26 GB |
| 81x81x10 | 241.3 | 2.0 | 117.5 | 27.9 | 14.3 | 13.9 | 67.3 | -1.6 | 102 GB |
| 121x121x10 | 623.7 | 4.7 | 292.2 | 108.3 | 30.2 | 32.8 | 170.5 | -15.0 | 227 GB |
| 161x161x10 | 1472.1 | 11.3 | 740.3 | 284.4 | 51.6 | 62.4 | 336.1 | -14.1 | 402 GB |

**Table 1.** Additional computation time needed for the calculation of Local Fractions in different settings, expressed in % in comparison to the total time needed when calculation of Local Fractions is not included. The first column shows the dimensions of the local region. The second column shows the total additional time required. Column three to eight show the breakdown of those fractions in the different subroutines ("comm." stands for communication time between compute nodes). "other" show the difference between total time and its components (it is principally due to uncontrolled differences in speed of the different compute nodes). The last column shows the additional memory required in total; it has to be multiplied by the number of species or sectors requested. The total time without calculation of Local Fractions in our test was 553 seconds.

The calculation of the Local Fractions only needs information from the nearest neighbors, see Eq. (9) and is therefore well suited for parallel processing in a space partition framework. While storing all the Local Fractions is memory demanding, the
data are distributed among the compute nodes, so that the memory requirement can be met by increasing the number of nodes.

In order to illustrate the computational cost, we can consider a typical model run, on a 400×260×20 grid (0.3 degrees longitude × 0.2 degrees latitude resolution), over one month (March 2016) on 160 processors that takes 553 seconds without the Local Fraction calculations. Table 1 shows the additional computational cost for computing the Local Fractions in our implementation. The mathematical operations required to compute the Local Fractions are proportional to the number of
sources considered and the size of the local area. In our implementation the additional time required for advection and diffusion grows faster, because of sub-optimal utilization of cache memory. Each of the process described in Sect. 2 requires only a few simple mathematical operations on each element of the LF array. The emission part only require to modify the lowest levels of the array (if emissions are restricted to them). The advection process has to be accounted for for each of the three dimensions, and is therefore more costly.

If one is only interested in the nearby sources (within a city, for example), the Local Fractions can be calculated at almost no additional cost. Remote sources can still be described, but at an additional cost.

A substantial amount of time can be required for writing results to disk, if all results are required at finer time resolution, for example every hour (in Table 1 the results are only written out once). This is mainly due to the large amount of data collected; for instance, for a local region of size 81×81×10 and for each sector or species, $400 \cdot 260 \cdot 81 \cdot 81$ values = 2.5 GB of data
have to be written to disk each time it is requested (only one vertical level is written out). The corresponding memory demand is calculated in the same way, but must further be multiplied by the number of vertical levels of the local region, then by two

because one array is needed to store the instantaneous values and one for accumulating the values over time, and multiplied by another factor two because the calculations are done in double precision.

## 5  Discussion

Local Fractions are a new concept that can help understand and analyse the origin of primary pollutants. It has the potential to be developed further, and a new range of applications is still being developed.

Compared to other approaches, there are always trade offs. The present method cannot at present describe non-linearities. That excludes all study of Ozone. Long range transport will also become unpractical at some point; although this is not inherent to the method and could be implemented in the near future.

### 5.1  Source apportionment

Source receptor relationships can be produced for any source and receptor within a region around the source. The size of this region can be chosen to be relatively large (100 grid cells or more). Since the fluxes are given from and to individual grid cells, small regions (typically cities) can be studied simply by adding up individual grid cell contributions. These small regions do not have to be predefined in the model simulations. Indeed, the relative contributions of sources that contribute to the pollutants within a city covering several grid cells can be determined in a post-processing step, using graphical user interfaces where the

user can choose the source region and source categories interactively.

Still, the method provides information about transport within a limited region only (the 'local region'). The choice of the size of this region is a balance between the computational cost and the distance to the sources of interest. For the study of a city, it may be sufficient to include a region covering the agglomeration. The total pollutants from sources outside of the local region are still quantified but without specification of their location, using the method presented in Sect. 3.4.

If the goal is to provide source-receptor matrices for large regions (countries), then this method is probably not appropriate in its present form as the computational cost may be too high, and the level of detail provided is not needed. For such an application the method should be modified, so that the tracking is not done for individual grid cells, but for larger source areas or group of emission sources.

### 5.2  Downscaling

One obstacle to combine fine scale (urban) and regional modelling is the problem of "double counting". In the regional scale model, there is usually only one total concentration value, without distinction between its origins. Distinguishing between urban and background pollution can be difficult in practice (Thunis , 2018).

Ideally, the regional model should only compute the background/regional contributions and the fine scale model can then add the local contribution. In a city, scales down to street level may be required. Those very fine scale models will not compute

accurately the transport between distant streets within the city and the regional model must account for those. But if the same emissions source are included both in the regional and fine scale model they will be accounted for twice.

The Local Fractions can give the relative contributions from different sources directly. Thus, it is possible to either redistribute or replace only the appropriate local contributions using the more accurate fine scale model.

An example for an operational downscaling tool is "uEMEP" (= urban EMEP), which combines the method described in this paper with the EMEP MSC-W air quality model (Simpson et al. , 2012), to provide daily air quality forecasts for all of Norway (https://luftkvalitet.miljostatus.no/ , Denby et al. , 2020).

## 5.3 Improved modelling

Concentration of pollutants near the surface are required to assess health impacts or dry deposition. However, in many CTMs, the lowest layer is several tens of meters thick, and the concentrations of pollutants will have a non-constant vertical profile within the layer. The shape of the profile will depend on the local conditions: if the pollutants are emitted locally at the surface the concentration will typically decrease with height, while the opposite is true for background pollutants. With the knowledge of the Local Fractions it is possible to improve the description of the vertical profile, and thus a more accurate estimation of, for instance, 3 meter concentrations (useful for health impact studies) or dry deposition rates.

As shown in Fig. 1 and 2 the Local Fractions vary strongly in space and time. If this information can be used to give better estimations of vertical profiles of pollutants it should have a significant effect on the results.

## 5.4 Future work

In this work, sources are always defined in an individual grid cell. The relative position of the source, $(\Delta x_s, \Delta y_s)$, could be replaced by a generic index that would point to more general groups of grid cells or regions. The formalism would be the same, except that emissions from any grid cell from the relevant region should be added together in the Local Fraction. This would allow for instance to distinguish individually all grid cells in the immediate vicinity of the receptor grid cell, and successively larger regions as the distance increases. Another application could be to define countries as emitter regions.

In the future we plan to generalize the method to also include chemical processes in some simplified form. The ambition is to still provide information for a very large number of sources, but to describe chemical processes in an approximate way. Compared to existing tagging methods, it will trade accuracy for computational efficiency.

*Code and data availability.* The full EMEP MSC-W model code and main input data are publicly available through a GitHub repository under a GNU General Public License v3.0 (name emep-ctm). The routines related to the Local Fractions are part of the standard model. The exact version of the model used to produce the illustrative examples used in this paper (rv4.33) is archived on Zenodo (doi: 10.5281/zenodo.3265912).

*Competing interests.* The authors declare that they have no conflict of interest.

*Acknowledgements.* This work has been supported by the AirQuip project funded by the Research Council of Norway (Project Number: 267734). The computations were performed on resources provided by UNINETT Sigma2 - the National Infrastructure for High Performance Computing and Data Storage in Norway.

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
