# Peer review of "Local Fractions - a method for the calculation of local source contributions to air pollution, illustrated by examples using the EMEP MSC-W model (rv4\_33)"

_Geoscientific Model Development, 2019_

## Referee Comment (RC1) · Anonymous Referee #1 · 16 Dec 2019

The manuscript of Wind et al. (gmd-2019-296) presents a new method for source apportionment called local fractions. The authors present the general idea of the method and show some results of it, implemented in the EMEP model. Further, a discussion about potential future applications is presented.

General comments:

[Figure]

The local fractions are a very interesting method, which is complementing to other methods for source apportionment. The general idea of the method is described well and the manuscript clearly fits into the scope of GMD. Before publication, however, I think major revisions are necessary. As the method is completely new the authors should characterise the method in more detail and answer questions like ''What window sizes are needed for which scientific questions?' Further, the authors need to discuss the pros and cons of their method compared to other methods in more detail and add details of the implementation of the method and the model set-up. In addition, the authors seem to overstate their method in some parts. More details are given in the specific comments below.

Specific comments:

- First of all the authors claim that their method is inexpensive wrt. the computational costs. Considering Table 1 I agree that the costs are low for small window sizes (e.g. 11x11 or 21x21), but with larger sizes the costs get remarkably large. Of course a simulation with 161x161 would replace 161x161 perturbation simulations (if I understand the method correctly), but for source apportionment studies one would usually not be interested in the contribution from 161x161 gridboxes to each other. Instead, one would be more interested in contributions from different regions/and/or emission sectors, right? For linear species this could be achieved by "just" adding additional tracers for different regions/emission sectors which should end up in far less additional costs compared to the large overhead for a 161x161 window. To better judge the the advantages of the method the authors should make clear how large the windows (x,y and z direction) for different purposes need to be. For the downscaling example given in Sect. 4.2 or the example in Sect. 4.3 small windows might be enough, but for the source apportionment I guess that very large windows are necessary. At least from the information of Figs 5,6,7 and 8 I have the feeling that 9 levels and a window of 80x80 are

necessary (at minimum) to account for all sources (e.g. a completeness of ≈ 1). Further, pollutants with different lifetimes should need different window sizes (for a completeness of 1, see Fig.4). I think is is very important that the authors characterise the method in more detail and critically discuss the advantages and disadvantages of their method compared to other methods for different scientific questions.

- In the description of the method the authors explain how they deal with emissions, advection, diffusion, deposition and chemistry, but it remains unclear how to deal with convection. On the coarse grid applied in the example convection needs to be parametrized I guess (providing convective mass fluxes). Therefore, please describe how you deal with convection in the LF framework.

- The presentation of the examples is rather vague and many details are missing. Clearly, the main goal of the authors is the description of the method in general (e.g., the mathematical concept), which is fine. However, the results depends heavily on the implementation of the method in one specific model and the authors give no details about the implementation procedure in EMEP MSC-W. The authors claim that *'The updates of the Local Fractions can be added on top of an existing model in separate routines.'* (p14l7ff) I am not familiar with the EMEP model, but when inspecting the provided code it seems to me that several modules (including the advection module) needed to be touched to implement the method. Therefore, I would strongly recommend that the authors add more details about the technical implementation in the model (maybe as Supplement). In this context I ask the authors to please provide examples of CTMs/CCMs where the flux at each grid box is an available quantity. To my opinion, the flux might not be an easily available quantity in many models (depending heavily on the applied advection scheme).

- The authors are not providing any details of the model set-up. Of course the

manuscript is not intended as model evaluation etc., but some basic details of the set-up would help to understand the presented results and it would help other users of EMEP to reproduce the results.

- The method assumes linearity for chemical processes, which is acknowledged by the authors. However, the authors state in the abstract that the method is valid for all primary pollutants, even tough some of them have non-linear chemistry (e.g. NOx, NO3). This should be clarified in the abstract. Further, the authors discuss in Sect. 4.4 that they aim to include chemical processes. The consideration of the full chemistry, however, would lead to a highly more complex implementation (and more costly wrt to the computational resources). In this case 'traditional' tagging approaches (e.g. Li et al., 2012, Emmons et al., 2012, Kwok et al., 2015, Valverde et al., 2016, Grewe et al., 2017, Butler et al. 2018 (and references therein)) might be superior compared to the LF approach. Could the authors please comment on this?

- On p14l11ff & p15l3 the authors mention the large memory footprint of the method. Information about the relative increase of the memory demand caused by the LFs would be interesting. How much more nodes are needed if the method is applied?

- The definition of 's' is unclear in the manuscript. On p3l7 's' is defined as the source term (i.e. emission sector). On p6l25 's' is defined as pollutant. In all example there is no differentiation between emission sectors an I assume that the number of emission sectors is 1 in all examples? Information like this should be part of the model description Section.

- Please reconsider the usage of the term 'contribution'. Especially in the introduction (p2l11ff) 'contributions' and 'impacts' are mixed. With the 'direct' method only impacts can be calculated and only for linear species 'impact' equals 'contribution' (see e.g., Thunis et al., 2019).

Technical corrections:

- The colour bars in Figure 4 are not very helpful. They are showing many orders of magnitude, which make it hard to judge the difference between direct and LF. Maybe additional difference plots could be provided additionally.

- Please check the references. For example Emmons et al., 2012 is listed in the bibliography but the reference seems to be missing in the manuscript. Further, I do not see how a reference to a manuscript in preparation is useful in any way (Denby et al.)

- p2l20f: Why using present perfect here? I would say the tagging method is unpractical in these cases.

- Caption figure 2: Please clarify what total emissions averaged over on month are. Are you showing total emissions of the month or an monthly average flux of molecules?

**Bibliography**

Butler, T., Lupascu, A., Coates, J., and Zhu, S.: TOAST 1.0: Tropospheric Ozone Attribution of Sources with Tagging for CESM 1.2.2, Geosci. Model Dev., 11, 2825-2840, https://doi.org/10.5194/gmd-11-2825-2018, 2018.

Emmons, L. K., Hess, P. G., Lamarque, J.-F., and Pfister, G. G.: Tagged ozone mechanism for MOZART-4, CAM-chem and other chemical transport models, Geosci. Model Dev., 5, 1531- 1542, https://doi.org/10.5194/gmd-5-1531-2012, 2012.

Grewe, V., Tsati, E., Mertens, M., Frömming, C., and Jöckel, P.: Contribution of emissions to concentrations: the TAGGING 1.0 submodel based on the Modular Earth Submodel System (MESSy 2.52), Geosci. Model Dev., 10, 2615-2633, https://doi.org/10.5194/gmd-10-2615- 2017, https://www.geosci-model-dev.net/10/2615/2017/, 2017.

Kwok, R. H. F., Baker, K. R., Napelenok, S. L., and Tonnesen, G. S.: Photochemical grid model implementation and application of VOC, NOx , and O3 source apportionment, Geosci. Model Dev., 8, 99-114, https://doi.org/10.5194/gmd-8-99-2015, http://www. geosci-model-dev.net/8/99/2015/, 2015.

Li, Y., Lau, A. K.-H., Fung, J. C.-H., Zheng, J. Y., Zhong, L. J., and Louie, P. K. K.: Ozone source apportionment (OSAT) to differentiate local regional and super-regional source contributions in the Pearl River Delta region, China, Journal of Geophysical Research: Atmospheres, 117, https://doi.org/10.1029/2011JD017340, http://dx.doi.org/10.1029/2011JD017340, d15305, 2012.

Thunis, P., Clappier, A., Tarrason, L., Monteiro, A. , Pisoni, E., Wesseling, J, Belis, C.A., Pirovano, G., Janssen, S., Guerreiro, C., Peduzzi, E.: Source apportionment to support air quality planning: Strengths and weaknesses of existing approaches. Environment International, 130, 2019
* * *

---

## Referee Comment (RC2) · Anonymous Referee #2 · 16 Dec 2019

Wind et al. describe a method for source attribution of primary particulate matter in the EMEP MSC-W model which they call "local fractions". Source attribution of air pollution is a useful model diagnostic and has important policy applications, for example in the calculation of source/receptor relationships. Several different methods for source attribution of air pollution exist already, but in each case different trade-offs must be made in order to keep the required computational resources within reasonable bounds. The method of local fractions appears to make a different set of trade-offs compared with other source attribution methods, and so therefore appears to make a unique

contribution to the air pollution modelling literature. So far it only seems that the method is useful for source attribution of primary PM, but nevertheless I think the manuscript falls within the scope of GMD.

I have a number of major concerns with the manuscript in its current form, which must be addressed before it could be reconsidered for publication. Of primary concern is the use of sometimes extremely vague language throughout the manuscript, which makes it difficult to follow exactly what the authors are describing. I also think the authors can do a much better job of explaining how their work fits in the context of other source attribution methodologies. The authors are quick to point out the advantages of their method, but any reader not already familiar with the existing literature will not understand the significance of the advantages and disadvantages of the local fractions method. Some additional context would help a lot here.

Specific comments

Page 1, line 4: "distinguish a large" should probably be "distinguish the contribution of a large".

Page 2, line 17: The references given here are not a good representation of studies which trace pollutant concentration back to emitted species by tagging (with or without consideration of nonlinearities). For PM, the study of Kranenburg et al. (2013) should be mentioned here. It is mentioned later in the manuscriupt, but should also be discussed here because it is a tagging method. For ozone, both Wang et al. (1998) and Wu et al. (2011) actually avoid the nonlinearities by tagging ozone based on its geographical region of formation, rather than the geographical region in which precursors are emitted. The reference to Grewe et al. (2013) is more appropriate here, since it is capable of attributing ozone to its emissions, but the authors could consider instead referencing the most up-to-date version of this method, as described by Grewe et al. (2017) and an actual application of this method for ozone attribution by Mertens et al. (2018). Alternative approaches also exist, which make different trade-offs. The approaches shared by Dunker et al. (2002) and Kwok et al. (2015), which tag ozone based on the chemical regime are both well-established and should be referenced in any discussion of modelled source attribution. Yet another approach was described recently by Butler et al. (2018) and has been applied by Lupascu et al. (2019).

The authors are right that large number of tracers can rapidly make tagging computationally expensive, but they should also point out that techniques exist to keep these problems in check. For example Butler et al. (2018) Restrict the number of tracers to a carefully chosen set of representative source sectors; Grewe et al. (2017) make use of the concept of "chemical families" to keep the number of tagged species within reasonable limits; and Lupascu et al. (2019) restrict the length of their simulation period to focus on a pollution episode of interest. Each of these approaches brings different trade-offs, but in each case also significant advantages: the ability to perform source attribution for secondary pollutants; and the ability to perform source attribution for long-range transport. These trade-offs are especially interesting in the context of the present manuscript, since one of the major ways in which the computational complexity of the local fractions method is kept computationally simple is by restricting the size of the "local region" for which the source attribution is performed.

Page 2, line 33: "can be built ... relatively easily" is a vague statement. More detail is needed here.

Page 3, line 5: An important detail missing here is that origins of the pollutants being tracked must be restricted to emissions within a "local region". This should be made clear up front, rather than making the reader wait 2 more sub-sections to find this out.

Page 3, line 8: "source regions" is very vague here. It would help the reader to know already at this stage that the present implementation considers each grid cell as a separate source region, but that in principle the method can be expanded to work with larger source regions.

Page 4, lines 7-8: "reasonable" cost and "preset" numbers of grid cells are used very

vaguely here. These terms are discussed later in the manuscript, but most readers would benefit from forward references to the relevant sections here.

Page 4, lines 12-13: "usually not necessary..." is very vague here. The authors show later that in fact extending vertical resolution to at least the height of the PBL is useful. The authors should also note that this also applies only to the pollutants they assess in their manuscript. Transport in the free troposphere is important for some pollutants such as ozone.

Page 4, line 17: Keeping the size of the LF array down to a reasonable size appears to be the main trade-off associated with this method, and this should be acknowledged here.

Page 4, line 18: The concept of the "local region" is first used here, and only implicitly defined by its context. It would help most readers tremendously if this concept could be introduced a lot earlier, with the explanation that setting the size of the local region represents the major trade-off with using this method.

Page 5, line 20: This is a good point, and could perhaps be mentioned earlier in the manuscript where the authors make the claim that their method is easy to implement in different CTMs.

Page 6, lines 8-9: Generally throughout the manuscript it would also be nice to have some discussion of the limitations of the method.

Page 6, line 27: "given distance" is very vague here. This is why it would be good to already have a well defined and discussed concept of what the "local region" is and why it is needed in this method.

Page 8, line 7: There is no justification given here for choosing 8 levels.

Page 8, lines 11-12: More detail is needed here. Why exactly is this "not a problem" and "actually an advantage" compared with the direct method?

Figures 5 and 6: Labels are missing for the x-axes. It would also be better to reverse the vertical ordering of the line color legends so that they correspond with the ordering of the lines in the plots.

Page 11, line 11: Does this mean that more vertical levels would be required when simulating summer months?

Figure 7: Here the size of the local region is referred to by its "distance", whereas everywhere else in the paper the actual size of the local region in grid cells is given. I presume that a distance of 20 is the same as a local region of 41x41, but this is not at all clear. Please use a consistent way of describing the size of the local region.

Page 14, lines 18-19: This seems like speculation (sub-optimal use of cache memory). Another possible explanation is that the extra memory requirements could be leading to increased communication overhead.

Page 15, line 1: Is this "substantial amount of time" already included in Table 1, or is this additional time? Can the authors quantify this?

Page 15, lines 2-3: The example given (local region of size 21x21x1) is not used anywhere else in the manuscript. It would be more useful to know about the extra storage requirements of the configurations which are actually evaluated in the manuscript. The authors should expand Table 1 to also include the extra storage requirements of the configurations given in this Table.

Page 15, line 5: "origin of pollutants" should acknowledge the limitations of the method. Based on the evaluation presented by the authors, it seems that this method can currently only analyse the origin of some kinds of primary pollutants.

Page 15, lines 11-13: The authors have not provided any other details about interactive graphical user interfaces. Is this something for future work? Or can the authors already provide a reference for this?

Page 15, line 17: This is an important point, and not necessarily a disadvantage of

the method. For some applications it may be acceptable to simply know that a certain amount of pollution originates from outside the local region. This provides some justification for other trade-offs which are made when using this method.

Page 15, line 19: Can the authors go into more detail about the "double counting" problem and how their approach solves it?

Page 15, lines 23-24: Which of the "several" problems are avoided and how? This text is way too vague.

Page 16, line 14: It seems to me that the local fractions deliver information about contributions, not sensitivities.

Page 16, line 14: Why wouldn't the local fractions add up to 100%, and why isn't this a problem? It seems that the final sentence of the manuscript creates all sorts of problems for the interested reader. The authors could consider simply deleting this sentence, and merging the previous sentence with the previous paragraph.

References

Emmons et al. (2012) is in the reference list, but not cited in the text.

The reference list is also quite short for a paper on source attribution of modeled air pollution, which is a well-established field. Discussion of the suggested additional literature would help to place the current work in context, as described in more detail above.

Additional literature

Butler, T., Lupascu, A., Coates, J., and Zhu, S.: TOAST 1.0: Tropospheric Ozone Attribution of Sources with Tagging for CESM 1.2.2, Geosci. Model Dev., 11, 2825-2840, https://doi.org/10.5194/gmd-11-2825-2018, 2018.

Dunker, A., Yarwood, G., Ortmann, J., and Wilson, G.: Comparison of source apportionment and source sensitivity of ozone in a three-dimensional air quality model,

Environ. Sci. Technol., 36, 2953-2964, https://doi.org/10.1021/es011418f, 2002.

Grewe, V., Tsati, E., Mertens, M., Frömming, C., and Jöckel, P.: Contribution of emissions to concentrations: the TAGGING 1.0 submodel based on the Modular Earth Submodel System (MESSy 2.52), Geosci. Model Dev., 10, 2615-2633, https://doi.org/10.5194/gmd-10-2615-2017, 2017.

Kwok, R. H. F., Baker, K. R., Napelenok, S. L., and Tonnesen, G. S.: Photochemical grid model implementation and application of VOC, NOx, and O3 source apportionment, Geosci. Model Dev., 8, 99-114, https://doi.org/10.5194/gmd-8-99-2015, 2015.

Lupascu, A. and Butler, T.: Source attribution of European surface O3 using a tagged O3 mechanism, Atmos. Chem. Phys., 19, 14535-14558, https://doi.org/10.5194/acp-19-14535-2019, 2019.

Mertens, M., Grewe, V., Rieger, V. S., and Jöckel, P.: Revisiting the contribution of land transport and shipping emissions to tropospheric ozone, Atmos. Chem. Phys., 18, 5567-5588, https://doi.org/10.5194/acp-18-5567-2018, 2018.
* * *

---

## Author Comment (AC1) · 29 Jan 2020

We thank the two referees for their constructive and pertinent comments.
We have now significantly extended the level of detail in the text, and have been more explicit on different points, like the requirements for an implementation in a new software and the tool's limitation to linear chemistry.
Below is a point by point answer to the specific comments. In blue are our comments, in black the original referee comments (first RC1, then RC2, and the revised manuscript with colors showing the modifications at the end).

Specific comments RC1

• First of all the authors claim that their method is inexpensive wrt. the computational costs. Considering Table 1 I agree that the costs are low for small window sizes (e.g. 11x11 or 21x21), but with larger sizes the costs get remarkably large. Of course a simulation with 161x161 would replace 161x161 perturbation simulations (if I understand the method correctly), but for source apportionment studies one would usually not be interested in the contribution from 161x161 gridboxes to each other. Instead, one would be more interested in contributions from different regions/and/or emission sectors, right? For linear species this could be achieved by "just" adding additional tracers for different regions/emission sectors which should end up in far less additional costs compared to the large overhead for a 161x161 window. To better judge the the advantages of the method the authors should make clear how large the windows (x,y and z direction) for different purposes need to be. For the downscaling example given in Sect. 4.2 or the example in Sect. 4.3 small windows might be enough, but for the source apportionment I guess that very large windows are necessary. At least from the information of Figs 5,6,7 and 8 I have the feeling that 9 levels and a window of 80x80 are necessary (at minimum) to account for all sources (e.g. a completeness of ≈ 1). Further, pollutants with different lifetimes should need different window sizes (for a completeness of 1, see Fig.4). I think is is very important that the authors characterise the method in more detail and critically discuss the advantages and disadvantages of their method compared to other methods for different scientific questions.
Yes, the method may indeed be very expensive for some applications. We have tried to be more careful about our claims. If the main goal is to get country-to-country source receptor matrices, then it may not be the right tool. We thus have added in section 5 "If the goal is to provide source-receptor matrices for large regions (countries), then this method is probably not appropriate in its present form as the computational cost may be too high, and the level of detail provided is not needed. For such an application the method should be modified, so that the tracking is not done for individual grid cells, but for larger source areas or group of emission sources."

• In the description of the method the authors explain how they deal with emissions, advection, diffusion, deposition and chemistry, but it remains unclear how to deal with convection. On the coarse grid applied in the example convection needs to be parametrized I guess (providing convective mass fluxes). Therefore, please describe how you deal with convection in the LF framework.
Yes, this should be commented. In section 2.3 we have now added
"For convection the same procedure can be used by replacing the diffusion operator in Eq. (11) by the convection operator . In the current EMEP MSC-W model version, the convective processes are not implemented in the Local Fraction calculations."
And further in section 3:
"The standard settings of the model do not include convection over Europe."

• The presentation of the examples is rather vague and many details are missing. Clearly, the main goal of the authors is the description of the method in general (e.g., the mathematical concept), which is fine. However, the results depends heavily on the implementation of the method in one specific model and the authors give no details about the implementation procedure in EMEP MSC-W. The authors claim that 'The updates of the Local Fractions can be added on top of an existing model in separate routines.' (p14l7ff) I am not familiar with the EMEP model, but when inspecting the provided code it seems to me that several modules (including the advection module) needed to be touched to implement the method. Therefore, I would strongly recommend that the authors add more details about the technical implementation in the model (maybe as Supplement). In this context I ask the authors to please provide examples of CTMs/CCMs where the flux at each grid box is an available quantity. To my opinion, the flux might not be an easily available quantity in many models (depending heavily on the applied advection scheme).

We have added section 4 with an explicit description of what is needed from an implementation point of view.

• The authors are not providing any details of the model set-up. Of course the manuscript is not intended as model evaluation etc., but some basic details of the set-up would help to understand the presented results and it would help other users of EMEP to reproduce the results.

We added "The parameter settings are essentially the same as what is used for the official EMEP-MSCW runs, using "TNO_MACC-III" emissions (2015 update of (Kuenen et al., 2014)). However to simplify the interpretation of the results, two important modifications have been introduced: a simplified advection scheme is used (see Sec. 3.2), and all emissions are released at the lowest level."

• The method assumes linearity for chemical processes, which is acknowledged by the authors. However, the authors state in the abstract that the method is valid for all primary pollutants, even tough some of them have non-linear chemistry (e.g. NOx, NO3). This should be clarified in the abstract. Further, the authors discuss in Sect. 4.4 that they aim to include chemical processes. The consideration of the full chemistry, however, would lead to a highly more complex implementation (and more costly wrt to the computational resources). In this case 'traditional' tagging approaches (e.g. Li et al., 2012, Emmons et al., 2012, Kwok et al., 2015, Valverde et al., 2016, Grewe et al., 2017, Butler et al. 2018 (and references therein)) might be superior compared to the LF approach. Could the authors please comment on this?

As mentioned in the abstract "non-linear chemical processes are not accounted for and only primary pollutants can be addressed", and the details of what the term "not accounted for" means for primary pollutants which are non-linear, is defined in section 2.5: the method will give some results, but the stronger the non-linear effects are, the less accurate the results will be. Also, in the intro we mentioned that "It will thus complement existing methods, but not replace them"

Nevertheless, we have modified the last section: "In the future we plan to generalize the method to also include chemical processes in some simplified form. The ambition is to still provide information for a very large number of sources, but to describe chemical processes in an approximate way. Compared to existing tagging methods, it will trade accuracy for computational efficiency."

• On p14l11ff & p15l3 the authors mention the large memory footprint of the method. Information about the relative increase of the memory demand caused by the LFs would be interesting. How much more nodes are needed if the method is applied?

We have added an entry in Table 1, showing memory demand. (The number of nodes

depends on the amount of memory available on each of them. 64 GB per node or more is typically available on modern HPC systems, meaning that the demand is not unreasonably large for the window sizes considered here)

• The definition of 's' is unclear in the manuscript. On p3l7 's' is defined as the source term (i.e. emission sector). On p6l25 's' is defined as pollutant. In all example there is no differentiation between emission sectors an I assume that the number of emission sectors is 1 in all examples? Information like this should be part of the model description Section.

Conceptually, as far as the Local Fraction is concerned, there are not much differences between sectors and pollutants. They are just different sources. In fact, our implementation does use the same index for describing both.

• Please reconsider the usage of the term 'contribution'. Especially in the introduction (p2l11ff) 'contributions' and 'impacts' are mixed. With the 'direct' method only impacts can be calculated and only for linear species 'impact' equals 'contribution' (see e.g., Thunis et al., 2019).

Modified to "difference"

Technical corrections:

• The colour bars in Figure 4 are not very helpful. They are showing many orders of magnitude, which make it hard to judge the difference between direct and LF. Maybe additional difference plots could be provided additionally.

Yes, but this large range makes it also difficult to plot meaningful relative or absolute errors. The point here is to show that the shape of the distribution looks the same, also for non-linear species. To quantify the limits would be interesting, but would require too many different test cases. This must be done in an other study.

• Please check the references. For example Emmons et al., 2012 is listed in the bibliography but the reference seems to be missing in the manuscript.

The list of references has been checked and revised

• Further, I do not see how a reference to a manuscript in preparation is useful in any way (Denby et al.). This is recommended by the journal ("Works "submitted to", "in preparation", "in review", or only available as preprint should also be included in the reference list"). We hope to be able to add more details in the reference before publication. Even without more details, it should help future readers to search for it, once both papers are published, hopefully.

• p2l20f: Why using present perfect here? I would say the tagging method is unpractical in these cases.

Modified to

"In cases where the number of different tagged sources is large this method becomes unpractical."

• Caption figure 2: Please clarify what total emissions averaged over on month are. Are you showing total emissions of the month or an monthly average flux of molecules?

This was not clear in the manuscript; wording now changed to "accumulated".

Specific comments RC2

Page 1, line 4: "distinguish a large" should probably be "distinguish the contribution of a large".

Yes, this has been changed.

Page 2, line 17: The references given here are not a good representation of studies which trace pollutant concentration back to emitted species by tagging (with or without consideration of nonlinearities). For PM, the study of Kranenburg et al. (2013) should be mentioned here. It is mentioned later in the manuscriupt, but should also be discussed here because it is a tagging method. For ozone, both Wang et al. (1998) and Wu et al. (2011) actually avoid the nonlinearities by tagging ozone based on its geographical region of formation, rather than the geographical region in which precursors are emitted. The reference to Grewe et al. (2013) is more appropriate here, since it is capable of attributing ozone to its emissions, but the authors could consider instead referencing the most up-to-date version of this method, as described by Grewe et al. (2017) and an actual application of this method for ozone attribution by Mertens et al. (2018). Alternative approaches also exist, which make different trade-offs. The approaches shared by Dunker et al. (2002) and Kwok et al. (2015), which tag ozone based on the chemical regime are both well-established and should be referenced in any discussion of modelled source attribution. Yet another approach was described recently by Butler et al. (2018) and has been applied by Lupascu et al. (2019). The authors are right that large number of tracers can rapidly make tagging computationally expensive, but they should also point out that techniques exist to keep these problems in check. For example Butler et al. (2018) Restrict the number of tracers to a carefully chosen set of representative source sectors; Grewe et al. (2017) make use of the concept of "chemical families" to keep the number of tagged species within reasonable limits; and Lupascu et al. (2019) restrict the length of their simulation period to focus on a pollution episode of interest. Each of these approaches brings different trade-offs, but in each case also significant advantages: the ability to perform source attribution for secondary pollutants; and the ability to perform source attribution for long-range transport. These trade-offs are especially interesting in the context of the present manuscript, since one of the major ways in which the computational complexity of the local fractions method is kept computationally simple is by restricting the size of the "local region" for which the source attribution is performed.

Thank you, we have included the recommended references.

However since our method only considers primary pollutants and exclude (at present) ozone, we prefer not to discuss in too much detail the additional challenges involved.

Yes, it is important to relate our method to other methods showing the trade offs involved and which type of applications where the Local Fraction method has its advantages (and the ones where it is not recommended). We have included a new paragraph in the Discussion section.

Page 2, line 33: "can be built ... relatively easily" is a vague statement. More detail is needed here.

We have added a new section about implementation with a more explicit description of what is needed and why we think it is "relatively easy to implement". In the introduction we have added "In Sect. 4 we will give an overview over what is required to implement the method in an existing CTM and discuss the performance in the EMEP MSC-W implementation."

Page 3, line 5: An important detail missing here is that origins of the pollutants being tracked must be restricted to emissions within a "local region". This should be made clear up front, rather than making the reader wait 2 more sub-sections to find this out.

Yes, we have added a new paragraph in the intro (see below)

Page 3, line 8: "source regions" is very vague here. It would help the reader to know already at this stage that the present implementation considers each grid cell as a separate source region, but that in principle the method can be expanded to work with larger source regions. Yes, we have added a new paragraph in the intro (see below)

Page 4, lines 7-8: "reasonable" cost and "preset" numbers of grid cells are used very vaguely here. These terms are discussed later in the manuscript, but most readers would benefit from forward references to the relevant sections here.
Yes, we have added a new paragraph in the intro (see below)

Page 4, lines 12-13: "usually not necessary..." is very vague here. The authors show later that in fact extending vertical resolution to at least the height of the PBL is useful. The authors should also note that this also applies only to the pollutants they assess in their manuscript. Transport in the free troposphere is important for some pollutants such as ozone.
Yes, we have added a new paragraph in the intro (see below)
Applications that need an explicit description of transport to the free troposphere (and back) would require more vertical levels. Therefore we have clarified that the size of the local region must be adapted to the specific needs of the application.

Page 4, line 17: Keeping the size of the LF array down to a reasonable size appears to be the main trade-off associated with this method, and this should be acknowledged here.
We have also added an entry in Table 1 that show quantitatively what this means.

Page 4, line 18: The concept of the "local region" is first used here, and only implicitly defined by its context. It would help most readers tremendously if this concept could be introduced a lot earlier, with the explanation that setting the size of the local region represents the major trade-off with using this method.
(Thank you, such things are difficult to realize for authors, but important for new readers!)
We have added a new paragraph in the introduction:

"In principle the method allows the definition of any group of sources, but here we will show results only for the case where each defined source is defined within a single grid cell. One key limitation, which makes the method manageable, is that the tagged values are stored only up to a preset horizontal and vertical distance from their source. We will call the region within this distance the "local region". The size of this region must be set as a balance between computational cost and the accuracy requirement of the application."

Page 5, line 20: This is a good point, and could perhaps be mentioned earlier in the manuscript where the authors make the claim that their method is easy to implement in different CTMs.

Page 6, lines 8-9: Generally throughout the manuscript it would also be nice to have some discussion of the limitations of the method.

Page 6, line 27: "given distance" is very vague here. This is why it would be good to already have a well defined and discussed concept of what the "local region" is and why it is needed in this method.
"Local region" is now defined in the introduction

Page 8, line 7: There is no justification given here for choosing 8 levels.
This is just an illustrative test that shows results under these conditions. (sensitivity to the vertical size of the window is shown in a subsequent figure)

Page 8, lines 11-12: More detail is needed here. Why exactly is this "not a problem"

and "actually an advantage" compared with the direct method?

We have tried to be more explicit: "The default fourth order scheme is slightly non-local, and the direct method would give spurious results very close to the sources; tracking and direct methods would give different results. For example, in the fourth order scheme, if emission are *reduced* in one gridcell, this can reduce the flux from the neighbouring grid cell in the upwind direction, thereby *increasing* the concentration of pollutants in the upwind grid cell.

This is however not a problem for the LF method (or any tracking method), and for short distances it is actually an advantage compared to the direct method."

Figures 5 and 6: Labels are missing for the x-axes. It would also be better to reverse the vertical ordering of the line color legends so that they correspond with the ordering of the lines in the plots.

Good idea, done

Page 11, line 11: Does this mean that more vertical levels would be required when simulating summer months?

Yes, that would be expected. The size of the local region must be adapted to the application. We also added in section 3: "The limitations of the method should be estimated for each concrete application. The examples in this section also provide methods for estimating different errors associated with the method (limitation of the size of the local region, non-linearities)."

Figure 7: Here the size of the local region is referred to by its "distance", whereas everywhere else in the paper the actual size of the local region in grid cells is given. I presume that a distance of 20 is the same as a local region of 41x41, but this is not at all clear. Please use a consistent way of describing the size of the local region.

Yes, done

Page 14, lines 18-19: This seems like speculation (sub-optimal use of cache memory). Another possible explanation is that the extra memory requirements could be leading to increased communication overhead.

It is more than speculation: we had access to more numbers than published. We have rerun these tests, and collected detailed timing information that is now presented in Table 1.

Page 15, line 1: Is this "substantial amount of time" already included in Table 1, or is this additional time? Can the authors quantify this?

The time for writing the data to disk, is (and was) included in Table 1 . We have rerun these tests, and a complete breakdown is now given.

Page 15, lines 2-3: The example given (local region of size 21x21x1) is not used anywhere else in the manuscript. It would be more useful to know about the extra storage requirements of the configurations which are actually evaluated in the manuscript. The authors should expand Table 1 to also include the extra storage requirements of the configurations given in this Table.

Done

"for instance, for a local region of size 81x81x10 and for each sector or species, 400*260*81*81$ values = 2.5 GB of data have to be written to disk each time it is requested (only one vertical level is written out). The corresponding memory demand is calculated in the same way, but must further be multiplied by the number of vertical levels of the local region, then by two because one array is needed to store the instantaneous values and one for accumulating the values over time, and multiplied by another factor two because the calculations are done in double precision."

Page 15, line 5: "origin of pollutants" should acknowledge the limitations of the method.

Based on the evaluation presented by the authors, it seems that this method can currently only analyse the origin of some kinds of primary pollutants.

Added the word "primary"

Page 15, lines 11-13: The authors have not provided any other details about interactive graphical user interfaces. Is this something for future work? Or can the authors already provide a reference for this?

We have a python script that works as a demo version. This is not meant to be the first version to be developed, but rather something basic that helps to look at the Local Fractions, as it is many dimensional and standard tools are not always adapted. We could put it on zenodo, together with a sample of data. Here is a temporary link (https://sandbox.zenodo.org/record/455127). If you recommend it, we can put it as permanent and reference it in the paper.
We have plans to make a more useful LF-GUI, but this has not concretized yet.

Page 15, line 17: This is an important point, and not necessarily a disadvantage of the method. For some applications it may be acceptable to simply know that a certain amount of pollution originates from outside the local region. This provides some justification for other trade-offs which are made when using this method.

Page 15, line 19: Can the authors go into more detail about the "double counting" problem and how their approach solves it?
Page 15, lines 23-24: Which of the "several" problems are avoided and how? This text is way too vague.

This section has been revised. Also added "In a city, scales down to street level may be required. Those very fine scale models will not compute accurately the transport between distant streets within the city and the regional model must account for those. But if the same emissions source are included both in the regional and fine scale model they will be accounted for twice."

Page 16, line 14: It seems to me that the local fractions deliver information about contributions, not sensitivities.
Page 16, line 14: Why wouldn't the local fractions add up to 100%, and why isn't this a problem? It seems that the final sentence of the manuscript creates all sorts of problems for the interested reader. The authors could consider simply deleting this sentence, and merging the previous sentence with the previous paragraph.

We really meant to say sensitivities. But we can agree that without further details, it does not give enough explanation. So rather than explaining the details of what we might do, we have deleted this sentence and replaced it by

[revised manuscript text omitted]

---

## Author Response (AR2)

**Answer to referee and annoted revised manuscript.**

(In black: referee comment, in blue our response)

Generally, the quality of the manuscript has improved. The limitations of the method are now much clearer. However, a little bit more work could have been done in providing a general discussion about needed window sizes for different purposes (see first comment from my initial review: ‚To better judge the the advantages of the method the authors should make clear how large the windows (x,y and z direction) for different purposes need to be'). Maybe the authors just don't agree with my comments, in this case they should just respond to the comment in and appropriate way.

We think the illustrative examples gives some answers of the level of accuracy that can be expected. In general we think we adressed this question at the beginning of section 3: "The Local Fractions will depend on a broad range of factors such as emission distributions, meteorological conditions, grid resolution, chemical regime, size of the local region etc. It is beyond the scope of this article to systematically quantify how all the possible situations affect the Local Fractions. The limitations of the method should be estimated for each concrete application. The examples in this section also provide methods for estimating different errors associated with the method (limitation of the size of the local region, non-linearities)."

In addition, a little bit more in depth discussion about the results in Table 1 would have been very insightful.

We have added: "Each of the process described in Sect. \ref{sec:method} requires only a few simple mathematical operations on each element of the LF array. The emission part only require to modify the lowest levels of the array (if emissions are restricted to them). The advection process has to be accounted for for each of the three dimensions, and is therefore more costly."

Minor comments:

If you use 's' as index for sector/pollutant please just define s at the beginning accordingly (s is the index defining [...], which can be either pollutant or sector). Otherwise this is just confusing to readers which are not familiar with the method (and readers of the manuscript usually are not familiar with the method)

We have reformulated the definition:

"$s$ is the index defining a pollutant or a pollutant from a specific sector. For example, $s$ can refer to primary particulate matter from any sector, or restricted to a power plant or the road traffic emissions in a specific source region."

Please revise the new part on p2l16-p2l30 completely, as it reads very confusing. Adding the term ‚differences' adds more confusion and does not help to clarify things here. Further, tagging describes only a method and is not limited to ozone molecules (see also p2l27ff)

Due to some repetition in the original version, the paragraph was confusing to read and therefore has been completely revised in the new version of the manuscript:

"Many chemical processes in the atmosphere are non-linear. For example, a doubling of the emission from one specific source will not necessarily double its contribution to air pollution levels. This also implies that the sum of contributions (from individual sources) calculated by the direct method (or by perturbation methods) will in general not be equal to the total air pollution level calculated in a simulation where emissions from all sources are included in full. Consequently, one has to distinguish between two different questions: 1) What is the effect of a

change in emissions from individual sources on air pollution? (air pollution sensitivity), and 2) What are the contributions of individual sources to air pollution? (source apportionment). Due to non-linearities, question 2 cannot be answered by reducing the emissions of individual sources to zero one-by-one. An alternative approach to estimate contributions from individual sources in model calculations is a technique known as "tagging", which distinguishes chemically identical molecules according to their sources. In the calculation the molecule is labelled (e.g. by a separate index) according to its source and then keeps this label during transport and chemical transformations. When analyzing air pollution levels within a given receptor area, the fractions of molecules with different labels can be considered separately, thereby giving an estimate of the contributions from the different sources. A series of methods have been proposed to address the contribution from different sources based on the 'tagging method' (e.g., Butler et al., 2018; Emmons et al., 2012; Dunker et al., 2002; Kwok et al., 2015; Grewe et al., 2013, 2017; Wang et al., 1998; Wu et al., 2011). Tagging methods are also useful for tracing primary pollutants (e.g., Kranenburg et al., 2013). However, in cases where the number of different tagged sources is large, the tagging methods can become excessively computationally expensive."

The authors did not answered my question about which models offer the advective fluxes as normal available quantities (p5l19f).
The short answer is that we don't know.
We were curious ourselves, and wanted to start to look at the code for Chimere, however it is not openly available, and we never got an answer to our online request
(https://www.lmd.polytechnique.fr/chimere/CW-download.php?page=CW-dwl-newuser.php)

P7l4: Where are the ‚official EMEP MSC-W model runs' (settings) documented?
The github versions are meant to be released with the setting for the official runs (i.e. the default settings are used). This is explained in the first lines of "doi: 10.5281/zen-odo.3265912": "The EMEP/MSC-W model version planned to be used on the EMEP status reporting of the year 2019 - rv4.33 - is released," with links to publications.

Technical comments:

P2L22: The sentence does not make sense. Please check.
Removed "address"

P15l13: Please clarify ‚in earlier versions'. I guess you meant earlier versions of EMEP?
We meant generally in the first versions in a development process, in the sense of "temporary version".  We have now reformulated as "in a simplified version".

[revised manuscript text omitted]